# Pharmacy Students’ Perceptions of Receiving Hands-On Continuous Glucose Monitoring (CGM) Education as Part of Their Core Curriculum: A Pre-Post Study

**DOI:** 10.3390/pharmacy13030078

**Published:** 2025-05-29

**Authors:** Joyce Y. Lee, Daniela Arcos, Daniella Chan, Celine Karabedian, José Mayorga

**Affiliations:** 1School of Pharmacy and Pharmaceutical Sciences, University of California, Irvine, CA 92697, USA; darcos1@uci.edu (D.A.); dychan3@hs.uci.edu (D.C.); ckarabed@hs.uci.edu (C.K.); jomayorga@altamed.org (J.M.); 2School of Medicine, University of California, Irvine, CA 92617, USA; 3AltaMed, Los Angeles, CA 90040, USA

**Keywords:** continuous glucose monitoring, pharmacy curriculum, hands-on education, diabetes, pharmacy students, active learning

## Abstract

Hands-on continuous glucose monitoring (CGM) training is yet to be integrated intentionally into most pharmacy curricula. The objective of this study was to examine pharmacy students’ perceptions of receiving hands-on CGM training as part of their core therapeutics in diabetes. Anonymous, voluntary pre-post surveys were administered to two cohorts of 3rd-year pharmacy students from a public pharmacy school in Southern California. Pharmacy students from other class levels were excluded. The surveys, consisting of Likert scale and open-ended questions related to different aspects of CGM, were administered via a web-based learning management system. Descriptive analyses were utilized to summarize the data. In total, 84 (98%) and 79 (92%) students completed the pre- and post-activity surveys, respectively, with an average of 12.5 days of CGM wear. After receiving the CGM training, 94% of the students recommended the integration of hands-on CGM training into the PharmD curriculum. In addition, the number of students who felt confident coaching patients on CGM use more than doubled from 30% to 85%, with 73% reporting added benefits of improved personal health behaviors. In conclusion, pharmacy students’ perceptions of integrating hands-on CGM education as part of their core curriculum were largely positive with added benefits of self-care.

## 1. Introduction

Diabetes impacts over 830 million people worldwide and poses significant health challenges, with a global economic burden of USD 1 trillion per year [1]. Uncontrolled diabetes also heightens the risk of various complications, including cardiovascular disease, nephropathy, neuropathy, retinopathy, and stroke [2,3]. In addition to medication optimization, effective diabetes management, facilitated by digital advancements such as continuous glucose monitoring (CGM), is essential in mitigating short- and long-term complications of diabetes [4,5,6,7]. CGM technology enables the real-time tracking of interstitial glucose levels, empowering patients and healthcare providers to augment treatment decisions without worrying about the frequency of uncomfortable and inconvenient finger pricks [4].

Since the inception of CGM, evidence has consistently proven that CGM leads to improved clinical outcomes, including decreased hemoglobin A1C levels, hypoglycemic incidents, and blood glucose variability [4,5]. In addition, in a retrospective analysis of administrative claims data of 571 patients with type 2 diabetes, the use of CGM was associated with a reduction in diabetes-related healthcare resource utilization costs, including inpatient hospital admission and length of hospital stay [6]. Furthermore, in a systematic review of 54 qualitative studies on patient experiences with CGM, CGM use was associated with improved patient diabetes self-management, confidence, and reassurance of life [7].

The effectiveness of CGM largely depends on patients receiving proper diabetes education, training, and support [7]. Pharmacists, who already play a pivotal role in diabetes self-management education and support, are increasingly involved in assisting patients with CGM acquirement, placement, interpretation, and troubleshooting. In a retrospective cohort study involving 315 adult patients with uncontrolled diabetes, pharmacist-led interventions resulted in significant reductions in A1C levels and hypoglycemia incidents compared to physician-led interventions, particularly among patients treated with insulin [2]. In addition, a 2023 scoping review of 249 studies highlighted the beneficial outcomes of pharmacist-driven CGM services, which included improved quality of life, patient empowerment, and glycemic control [8]. These studies underscore the critical role of pharmacists in improving diabetes outcomes through the implementation of innovative diabetes technologies.

As CGM becomes increasingly prevalent, pharmacists and new pharmacy graduates are called upon to assist patients in utilizing these devices and interpreting CGM data to enhance diabetes management. To maintain their crucial role in healthcare teams, pharmacists and new pharmacy graduates alike must stay up to date and practice-ready to assist their patients effectively [9]. In a study that examined the provision of CGM education in pharmacy programs in the US via an online survey distributed to 139 accredited US pharmacy programs [10], 51 out of 57 participating programs reported providing some form of CGM education through therapeutic courses, care labs, and electives. However, the median time spent was 1 hour, with only 16 programs (33.3%) providing hands-on CGM education. Lack of time was cited as the main reason among the programs that did not offer CGM education. Relatedly, it is also unclear if pharmacy students are ready to participate in CGM training. This study examined pharmacy students’ perceptions of receiving hands-on CGM education as part of their core curriculum.

## 2. Materials and Methods

### 2.1. Study Design and Setting

This was a pre-post study conducted among two cohorts of 3rd-year pharmacy students who received hands-on CGM training as part of their therapeutics course from September 2023 to December 2024, offered by a 4-year Doctor of Pharmacy Program from a public school of pharmacy and pharmaceutical sciences located in the Los Angeles-Orange County region of Southern California. The pharmacy program in this academic institution was founded in 2020 with the mission to be a driving force for advancing the health and wellness of individuals and society through innovative and integrative learner-centered education.

### 2.2. Participants

All third-year pharmacy students who registered successfully to take the fourth module in a series of pharmacotherapy core therapeutic courses were invited to participate. Pharmacy students of other grade levels or those who did not fulfill the prerequisites to take the specified pharmacotherapy core module were excluded.

### 2.3. Study Procedures and Outcomes

#### 2.3.1. Pre- and Post-Activity Surveys

The anonymous, voluntary pre-activity survey was developed by the course coordinator (JL), who is also a certified diabetes care and education specialist (CDCES). The survey consisted of five items with one open-ended question and four Likert scale questions. The open-ended question asked about the pharmacy students’ current working status in a pharmacy, and if relevant, pharmacy students were asked to further elaborate on their practice setting and the type of patients managed. Three Likert scale questions asked the pharmacy students to rank levels of agreement (0 = not applicable, 1 = strongly disagree, 2 = disagree, 3 = neutral, 4 = agree, and 5 = strongly agree) based on (1) the degree of anxiety pharmacy students have for the CGM hands-on education activity, (2) the importance of learning about CGM to prepare them to be a practice-ready pharmacist, and (3) the need to incorporate hands-on CGM training intentionally as part of their core curriculum. The fourth Likert scale question asked the pharmacy students to rank their understanding and knowledge of CGM before receiving the hands-on CGM education (0 = not applicable, 1 = no knowledge, 2 = some knowledge, 3 = good knowledge, 4 = very knowledgeable).

Part of the anonymous, voluntary post-activity survey was adapted and extracted from the Continuous Glucose Monitoring Satisfaction Scale (CGM-SAT) [11], where 11 questions out of the 44-item CGM-SAT questionnaire that focused on satisfaction with the emotional behavioral and cognitive effects of CGM use were included. The post-activity survey consisted of 16 items, which included 4 open-ended questions and 12 Likert scale questions. The open-ended questions asked the pharmacy students to (1) elaborate on the number of days of CGM wear, (2) best and (3) worst part about wearing a CGM over a duration of time, and (4) the relevance of intentionally integrating hands-on CGM education as part of their core pharmacy curriculum. The 12 Likert scale questions asked the pharmacy students to rate their levels of agreement (0 = not applicable, 1 = strongly agree, 2 = agree, 3 = neutral, 4 = disagree, and 5 = strongly disagree) for eight statements related to their experience wearing a CGM sensor, two statements on confidence and understanding of CGM use in diabetes, and two statements on enthusiasm for learning about CGM and the importance of incorporating hands-on CGM education as part of the core pharmacy curriculum.

#### 2.3.2. Hands-On CGM Education (Table 1)

The hands-on CGM activity was developed and integrated as part of a 7-unit therapeutics course, which covered topics related to endocrinology and gender health during the Fall Quarter of a 3rd-year pharmacy curriculum. Approximately 11.0 h of diabetes therapeutics lectures were delivered over two weeks (out of an 11-week quarter schedule), with the hands-on CGM education (Parts I and II) incorporated in the first and last lectures of the 2-week diabetes therapeutics lecture series. The diabetes therapeutics lecture series and the hands-on CGM education were delivered by the faculty course coordinator with an advanced diabetes credential (i.e., CDCES). Before receiving the hands-on CGM education, pharmacy students were asked to complete a set of self-guided reading assignments on (1) the evolution of diabetes technology [12], (2) the CGM instructional user guide specific to the CGM model the pharmacy students would use during the CGM training [13], and (3) a video link on step-by-step CGM placement [13] (Table 1).

**Table 1 pharmacy-13-00078-t001:** Sample schedule on incorporating CGM hands-on education activity into core diabetes therapeutics lecture series.

Learning Type	Duration	Activities
Self-study (outside class time) (1.0 h)		Review the article on the evolution of diabetes technology [12] Read the get-started sensor guide [13] Watch video on CGM sensor placement [13]
Lecture 1	CGM Education Part I (1.5 h)	Review self-study material Show-and-tell with one faculty member Hands-on CGM wear activity
	Diabetes therapeutics (1.0 h)	Diabetes overview Diabetes pharmacotherapy Diabetes complications Case studies
Lecture 2	Diabetes therapeutics (2.5 h)	
Lecture 3	Diabetes therapeutics (1.5 h)	
Lecture 4	Diabetes therapeutics (1.5 h)	
Lecture 5 *	CGM education Part II (1.0 h)	CGM wear experience follow-up, data interpretation review, open discussion, question and answer>
	Diabetes therapeutics (1.5 h)	Wrap up Case discussion with integrated CGM data interpretation

* Fourteen days after lecture 1 to ensure that CGM Education Part II was delivered upon the completion of CGM wear.

On the day of the hands-on CGM education (Part I), the pharmacy students were instructed to sit in groups of 5–6 per table while the faculty reviewed the self-study materials with the pharmacy students and delivered an interactive didactic lecture on CGM data interpretation using patient cases. The faculty then demonstrated the CGM sensor attachment through show-and-tell, followed by pharmacy students applying the sensor to their own arms. The sensors used by the students were Freestyle Libre supported by Abbott Diabetes Care Educational Grant for No Charge Product awarded to the course coordinator.

Part II of the hands-on CGM education was held during the last diabetes therapeutics lecture series (14 days after completing the sensor wear), where the course coordinator followed up with a discussion on CGM wear experience coupled with a review on the interpretations of sample ambulatory glucose profiles using different patient case scenarios developed by the course coordinator. The lecture also addressed challenges commonly encountered by patients wearing CGM sensors and aspects to look out for when coaching patients on the use of CGM. Furthermore, the pharmacy students were also given time to share their own CGM wear experiences.

#### 2.3.3. Study Workflow

The pre-activity survey was administered before the disbursement of the mandatory self-study materials. The post-activity survey was administered 14 days after the day of the hands-on CGM training (end of the FreeStyle Libre CGM wear). All survey items were administered using CANVAS, a web-based learning management system.

### 2.4. Statistical Analysis

Descriptive analyses were utilized to summarize the deidentified data from pre- and post-activity surveys. Categorical data statistics were presented as percentages and frequencies. Divided bar graphs were used to visualize and compare the number of pharmacy students who expressed different degrees of agreement or disagreement related to the experience of CGM sensor wear and their outlook toward the intentional integration of hands-on CGM education as part of their core pharmacy curriculum. Open-ended questions were analyzed thematically by two trained research assistants through the identifications of thematic patterns within the responses. For the thematic analysis, we took on the six-step process, which included response familiarization, coding, theme formation and evaluation, followed by defining and naming the themes [14].

## 3. Results

### 3.1. Pre-Activity Survey

In total, 84 (98%) pharmacy students completed the anonymous, voluntary pre-activity survey, of which 58 (69%) reported working in a pharmacy as an intern. Of these 58 students, the majority (39, 67%) reported working in an outpatient or inpatient pharmacy setting, while 19 (33%) did not provide information on the details of their pharmacy work.

Among those who completed the pre-activity survey (Figure 1), 71 (85%) agreed or strongly agreed that having a good understanding of CGM would better prepare them to become a pharmacist, while 5 (6%) felt indifferent, another 7 (8%) disagreed or strongly disagreed, and 1 (1%) did not provide a rating. When the pharmacy students were asked if CGM-related hands-on activity should be made compulsory in pharmacy education, 47 (56%) agreed or strongly agreed, 25 (30%) felt indifferent, 11 (13%) disagreed or strongly disagreed, and 1 (1%) did not provide a rating. Half of the students (44, 52%) reported having some knowledge about CGM before receiving the hands-on CGM education, while 25 (30%) reported having good or very good knowledge, 14 (17%) reported having no knowledge about CGM, and 1 (1%) did not provide a rating. When the student pharmacists were asked if they felt nervous about attaching a CGM sensor to their arm, 44 (52%) agreed or strongly agreed, 17 (20%) felt indifferent, 20 (24%) disagreed or strongly disagreed, and 3 (4%) did not provide a response or indicated that the question was not applicable.

### 3.2. Post-Activity Survey

In total, 79 (92%) pharmacy students answered the post-activity survey with 56 (71%) wearing the CGM sensor successfully for 14 days. Among the 23 (29%) students who failed to complete the 14-day CGM sensor wear due to sensor detachment, the average duration of CGM sensor wear was 6.8 days, ranging between 1 and 11 days. Overall, the average CGM sensor wear duration among all participating pharmacy students was 12.5 days.

With regard to the CGM experience (Figure 2), most participating pharmacy students who completed the post-activity survey agreed or strongly agreed that wearing a sensor did not affect how they were perceived by others (46, 58%) or caused embarrassment (61, 77%). Most participants disagreed or strongly disagreed that wearing a CGM sensor interfered with their daily activities (58, 73%). Relatedly, most participants either disagreed or strongly disagreed (46, 58%) or felt neutral (17, 22%) that wearing the sensor caused pain or discomfort. A total of 74 (94%) students felt that it is important to mandate CGM training as part of the pharmacy core curriculum, and 61 (77%) reported that they would wear the sensor again as part of their learning if given the opportunity. The majority of the pharmacy students reported that the use of CGM was not as complicated as they had imagined (65, 82%) and it helped them understand the topic of diabetes better (67, 85%). Furthermore, 67 (85%) felt more confident talking about CGM sensors to their peers or current/future patients. Aside from the learning experience, 58 students (73%) also reported that CGM training reminded them to stay healthy daily.

When the pharmacy students were asked to comment freely on the best things about participating in the CGM training, the most mentioned benefits were improved CGM understanding and patient counseling abilities (51%), followed by exploring one’s own sugar levels (30%), acquiring empathy (16%), and having first-hand experience with the device (11%). The worst things about the CGM training were largely mechanical in nature, with disruptive alarms being the most mentioned (24, 30%), followed by decreased sleep quality (18, 23%).

## 4. Discussion

This study examined pharmacy students’ perceptions of receiving hands-on CGM education as part of their core curriculum offered by a traditional 4-year postgraduate pharmacy school. The overall activity was delivered successfully by one faculty member without requiring additional administrative or pharmacist support during or outside the diabetes therapeutics lecture series. The pre- and post-activity surveys showed that hands-on CGM education not only improved students’ confidence in using CGM but also helped students’ overall understanding of diabetes. Furthermore, hands-on CGM education improved students’ awareness regarding their health and self-care practices beyond classroom learning. This study highlights the benefits of incorporating hands-on CGM education as part of a core pharmacy curriculum from the perspective of pharmacy students and provides insights into the importance of staying abreast of diabetes technology to ensure proficiency in managing the most prevalent chronic condition worldwide.

In this study, more than half of the pharmacy students felt nervous about applying the CGM sensors to their upper arm before receiving hands-on CGM education. After participating in the CGM training, 82% felt that applying the sensor was not as hard or as complicated as they had imagined. Moreover, the number of pharmacy students who felt confident talking to their patients and peers about CGM more than doubled to 85% after receiving hands-on CGM education. A cross-sectional study that evaluated injection-related anxiety among medical and pharmacy students reported that medical students were two times less likely to experience injection-related anxiety compared to pharmacy students [15]. This was attributed to the fact that medical students had more hands-on exposure to needles throughout their coursework. In addition, in other studies, including a meta-analysis of 15 studies that evaluated the effects of incorporating simulation and hands-on activities in training, nursing students confirmed that active learning not only improved self-confidence but also improved communication with patients and their team member [16,17,18]. Indeed, hands-on activity can close the gap between theory and practice, replacing anxiety with familiarity and confidence.

In this study, approximately 29% of pharmacy students experienced early CGM sensor detachment after wearing CGM for an average of 6.8 days. According to a study that evaluated the CGM adverse events database constructed by the U.S. Food and Drug Administration, the rate of detached CGM sensors in 2022 ranged between 1.4% and 9.2% [19]. While reasons for CGM sensor detachment were not surveyed among our pharmacy students, failure to remove any oily residue on the skin at the time of applying the CGM sensor has been reported to be one of the common reasons for CGM sensor detachment [20]. Future hands-on CGM training may wish to place additional emphasis on preparing the skin before applying the CGM sensor (e.g., clean with alcohol wipe thoroughly, avoid using body lotion, etc.) or supply optional medical-grade adhesive to those who may be at higher risk of CGM detachment (e.g., intense physical activity). Furthermore, approximately 30% of the pharmacy students reported disruption related to the CGM alarms, which contributed to 23% of participants reporting sleep interruption and unnecessary anxiety. A small behavioral intervention pilot that examined the sleep quality of parents with children with type 1 diabetes who used CGM also reported poor sleep quality due to device alarms and continuous worrying about hypoglycemia [21]. Therefore, in addition to coaching the students on the interpretation of CGM data, training on the technicality of the CGM devices such as their alarm mode (e.g., use of vibration, use of temporary silent mode) is also called for to minimize unnecessary worries or interruptions from the alarm.

Lastly, the number of pharmacy students who acknowledged the importance of making hands-on CGM education compulsory in the pharmacy curriculum increased from 56% to 94% after participating in the training. Relatedly, 77% of the pharmacy students reported interest in wearing the CGM sensor again if given the opportunity, and approximately 85% felt that incorporating hands-on CGM education into the diabetes therapeutics lecture series helped them learn about diabetes. In the United States, the American College of Clinical Pharmacy Pharmacotherapy Didactic Curriculum Toolkit continues to advocate for diabetes as a tier 1 topic that must be covered by all colleges [22]. In 2022, the American Diabetes Association expanded its recommendations for CGM use for people with any form of diabetes and a variety of insulin regimens [23]. The use of CGM to improve health behaviors in people with or without insulin dependence, as well as those with prediabetes, is also promising [24]. With CGM becoming the mainstay of diabetes monitoring, and traditional glucose meters becoming increasingly obsolete, pharmacists who serve as frontline patient advocates must keep abreast of diabetes knowledge and technology to ensure successful care delivery. Diabetes technology, therefore, is a crucial part of diabetes therapeutics that pharmacists must master to ensure the indispensable role of pharmacists in diabetes care, and early exposure to diabetes technology as pharmacy students will better prepare them to become practice-ready.

This study had several limitations. First, we only assessed pharmacy students’ perceptions of hands-on CGM education from one pharmacy school. While we were unable to account for possible response variability among different pharmacy schools, we have observed similar perceptions from two cohorts of 3rd-year pharmacy students consecutively since the inception of our pharmacy program in 2020, and with consistent positive findings that have set the precedent for incorporating hands-on CGM education into our pharmacy curriculum. Second, this was a pre-post study that did not provide statistically definitive conclusions about casual relationships. However, the pre-post study design provided us with practical insights into the value that hands-on CGM education can bring to our pharmacy students. Lastly, the student pharmacists only used one type of CGM sensor throughout the hands-on activity, limiting their exposure to other types of CGM sensors. Since CGM sensors share similar functionality, the didactic portion of the CGM education highlighted the main differences among different CGM sensors in the market to ensure a holistic learning experience. In addition to addressing these limitations, future studies may also compare student perceptions towards hands-on CGM education conducted at different time periods (e.g., prior to introductory pharmacy practice experiences vs. prior to advanced pharmacy practice experiences) and among pharmacy schools from different geographic locations.

## 5. Conclusions

Overall, the hands-on CGM training was well received by the pharmacy students. The training enhanced pharmacy students’ confidence in CGM counseling, and the intentional integration of hands-on CGM education as part of the core diabetes therapeutics also enhanced students’ understanding of diabetes with added benefits of self-care.

## Figures and Tables

**Figure 1 pharmacy-13-00078-f001:**
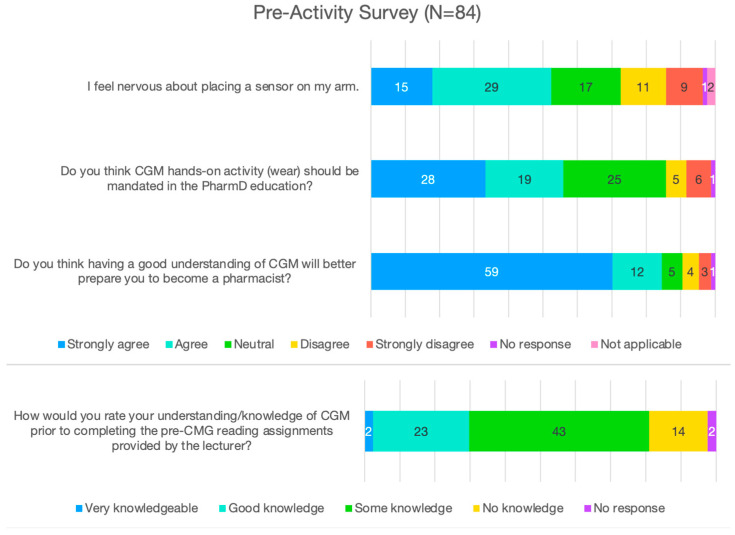
Pharmacy students’ pre-activity survey outcomes.

**Figure 2 pharmacy-13-00078-f002:**
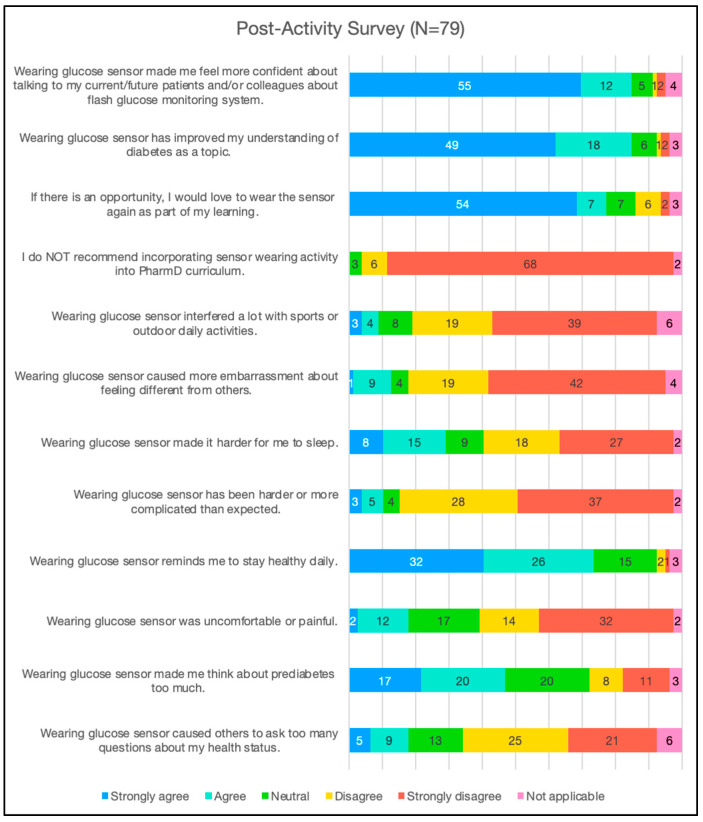
Pharmacy students’ post-activity survey outcomes.

## Data Availability

The authors have complete access to the anonymized study data and access is still ongoing. The materials supporting the findings are available. However, reader access is only available upon request and at the discretion of the corresponding author or the PI of this study.

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
