# Peer review of "Pharmacy Students’ Perceptions of Receiving Hands-On Continuous Glucose Monitoring (CGM) Education as Part of Their Core Curriculum: A Pre-Post Study"

_pharmacy, 2025, doi:10.3390/pharmacy13030078_

Round 1
Reviewer 1 Report
Comments and Suggestions for Authors
Journal Pharmacy (ISSN 2226-4787)
Manuscript ID pharmacy-3513925
Title Pharmacy Students’ Perceptions of Receiving Hands-on Continuous Glucose Monitoring (CGM) Education as Part of their Core Curriculum: A Pre-Post Study
Thank you for the opportunity to review this article describing an important topic in pharmacy practice and education. This was an interesting study but did not include results beyond descriptive statistics, which reduces the value as acknowledged in the limitations. However, the manuscript is well-written, and results indicated this activity was beneficial for students, so it is still an interesting manuscript. My feedback is below:
Abstract:
line 12: “An anonymous and voluntary pre-post surveys” reword
Background:
The background nicely describes why CGM education for student pharmacists is important. However, no information is provided about how this has been taught in pharmacy education or placement in the curriculum beyond mentioning citation 9. I suggest adding a paragraph describing how this topic is integrated in pharmacy education. Citation 9 includes lots of detailed information about how this is taught that could be incorporated.
Page 2 line 46: please add a citation for this statement.
Page 2 line 55: “these studies, among others…” please add citations for the others or remove this from the sentence. The two studies you list already make a strong argument so you may not need more.
Methods:
Sections 2.1 and 2.2 describe placement in curriculum and then 2.3 describes the survey. I suggest moving 2.3.2 up before the study procedures so it flows with the education part.
Results:
Figures missing axis titles.
The figures are difficult to digest. I can't think of a suggestion that would improve these but I believe the clarity of the results displayed can be improved.
Results are clearly conveyed.
Discussion:
First sentence statement about this being the first study does not seem correct. It looks like Creighton faculty have already conducted this but only presented as a poster. https://www.ajpe.org/article/S0002-9459(23)00319-4/fulltext. I suggest taking the opportunity to compare and contrast your results with their results from the poster abstract.
Discussion overall is very good and conveys the most interesting parts of the survey data.
Conclusion: I suggest adding a brief restatement of the results.
Author Response
Thank you for your comments. Please see attached.

Reviewer 2 Report
Comments and Suggestions for Authors
I congratulate the authors on their work!
REVIEW
This article aims to assess the impact of students' opinions on integrating continuous glucose monitoring (CGM) into the training curriculum. The study analysed anonymous surveys dispensed to third-year pharmacy students in a public pharmacy school in Southern California. The scope was to obtain an undergraduate’s view on gaining these types of abilities.
The manuscript is significant for the pharmaceutical education and training sector and is relevant nowadays when any additional practical skill adds value to the pharmaceutical act and benefits the patient. The references are recent publications relevant to the paper with very few self-citations.
The authors structured their study into the mandatory parts: introduction, materials and methods, results, discussions and conclusions. The references are recent publications relevant to the paper without self-citations.
In the paper, the authors have analysed the importance of CGM preparation from the student's point of view. The main questions addressed are: do they recognize its importance and do they feel confident in applying the acquired knowledge in day-to-day practice?
The CGM technique is a modern and relatively new technique that improves disease management, boosts confidence and improves the quality of life for diabetes patients.
The authors analyzed the responses from two groups of 3rd-year pharmacy students before and after receiving hands-on CGM training for 4 months.
As methods, they used two surveys: the pre-activity one which was anonymous, voluntary and consisted of five items and a post-activity survey including 16 items.
Data were analyzed using descriptive statistics, percentages and frecquencies. The study methodology is scientific, rigorous, and well-detailed.
The authors found that the majority of pharmacy students changed their view on the CGM technique:
- more than 80% felt it was not as hard or complicated as they had envisioned,
- 85% felt confident in sharing their experience and gained abilities helping patients and peers,
- the participants will no longer fear injection-related anxiety compared to medical and other pharmacy students (who didn’t participate in the survey),
- the trainees declared improved self-confidence and better communication skills with patients and their colleagues,
- the number of votes in favour of introducing this course into pharmacy training almost doubled (from 56% to 94%) after participating in training,
- the majority of participants (77%) agreed to wear the CGM sensor again if needed.
Only 29% experienced early CGM sensor detachment after almost 7 days of wearing. But that fact could be explained by not obeying the rules of utilization of the device.
Also, the results obtained indicated disruption relating to CGM alarms in 30% of cases which led to 23% of participants reporting sleep interruption and unnecessary anxiety. However, even if the optional alarms of many CGM sensors can be turned off, the fabrication setting concerning urgent low glucose alarms cannot be shut down, according to the researchers.
The results are reproducible with the methods used. The table and two figures showcase the data clearly and are easy to interpret and understand. Data were correctly interpreted across the manuscript.
The authors conclude that CGM training was approved by the pharmacy students. The conclusions are clear and derive from the obtained results; they are consistent with the evidence presented and address the main questions.
The main strengths of the paper are the following:
- it proposes a valid methodology for assessing student’s opinions concerning the modification of undergraduate pharmacy curricula,
- the fact that the questionnaire was anonymous supports the authenticity of the received answers,
- the thorough description of the data analysis methodology proves the correctitude of the research,
- the clear description of the proposed curricula,
- the minimum usage of faculty’s resources proves the efficiency of this training implementation,
- the analysis of the consequences of sensor detachment cases improves the quality of the paper.
As a weakness may be stated the fact that the authors have not presented (have not calculated ?!) the p-value which could underline the statistical significance of their results.
Just a small typing error on page 4, line 157 the word deidentified data will surely be corrected.
There are no ethical or data availability problems.
The paper is well written, clear, easy to read and has meaningful conclusions concerning the education and training of future professionals in pharmacy.
Author Response
Thank you for your response. Please see attached.

Reviewer 3 Report
Comments and Suggestions for Authors
The paper describes the incorporation of hands-on CGM education and the pharmacy students' views on this. The paper is well-written and of interested to those working in pharmacy education.
I have some minor comments:
Please add information in the method section how the answers from the open-ended questions were analysed. It appears from the results section that a quantification was made but this also needs to be described in the methods section.
In the conclusion it is mentioned that the students' understanding of diabetes was enhanced but this was not directly measured in the study. It is the students' perceptions that are investigated and their perceptions of an enhanced understanding of diabetes.
Author Response

(The authors gave the same response as above.)

Reviewer 4 Report
Comments and Suggestions for Authors
This study, titled “Pharmacy Students’ Perceptions of Receiving Hands-on CGM Education as Part of their Core Curriculum: A Pre-Post Study,” has a primary objective to examine third year pharmacy student perceptions of completing a CGM educational module which included an educational wear experience. This manuscript shares an example of how to successfully integrate CGM education within the core pharmacy curriculum, which is important to highlight since this is primarily done in elective courses in other schools, if at all. The educational intervention is well described and appears to be complete. I appreciate that the study includes data from two student cohorts to increase number of participants. While it is valuable to share the educational model used in incorporating CGM into the core curriculum, there are significant limitations to the study design itself. The pre/post survey is largely focused on user experience within the CGM educational wear, such as assessing frequency of occurrence of device challenges that users may experience, rather than focusing on effectiveness of the educational intervention. This does not align with the primary study objective. Pre/post survey questions regarding inclusion of an educational wear experience in the curriculum were not specific to core versus elective curriculum, and the questions asked were different pre/post, limiting comparability of the results. The study would be strengthened by the inclusion of a comparator group who did not receive this education intervention. It would also be strengthened by the inclusion of a knowledge assessment or comparison of performance on course assessments related to CGM education. Based on the challenges with study design, I do not find this manuscript suitable for publication. I would encourage the authors to revise the manuscript to focus on the education intervention, possibly as a commentary, rather than as a research study based on the limitations of the data.
Specific comments:
Page 1, lines 9-10: The abstract states CGM use “is yet to be integrated intentionally into most of the health professional curricula,” however the main text only speaks to the profession of pharmacy. Recommend changing to focus on pharmacy curricula in the abstract to align with the manuscript.
Page 1, line 33: It does not track blood glucose levels, recommend changing to interstitial glucose levels
Page 3 lines 104-105: Which questions were adapted from the Continuous Glucose Monitoring Scale specifically? What was adapted? Is this a validated tool?
Page 4, line 140: Which Libre devices were used? Were there any other asks of learners who wore the device, such as a living with diabetes simulation?
Results: Much of the text is repeated in the figures. Recommend editing the text to serve as a supplement to the figures rather than listing out the same results.
Figures 1 and 2: Review for typos and accuracy. The current formatting is a bit difficult to read. Could the numbers be bolded, or formatted differently, to improve readability? Recommend listing not applicable next to no response in the figure.
Discussion: The discussion aligns with the results of this study, however, neither are well-aligned with the study objective, which is to examine pharmacy students’ perceptions of receiving hands-on CGM training as part of their core therapeutics in diabetes. Alternatively, much of the discussion compares the student user wear experiences to those seen by people living with diabetes in real-world settings.
Page 8, line 223-224: Since this is based on a survey of students who felt their understanding of diabetes had improved rather than a knowledge assessment, I would recommend adjusting the text to ensure the outcomes are not overstated. What evidence supports their self-assessment that “wearing glucose sensor has improved my understanding of diabetes as a topic”?
Page 8 lines 235-237: This study reference appears out of place at present. What is the connection to the results of this study?
Page 9 lines 295-299: Based on this statement, is it really a limitation to only use one sensor? Did the authors anticipate that using different sensors would produce different results?
Author Response
Thank you for your response. Please see the attachment.

Reviewer 5 Report
Comments and Suggestions for Authors
The authors have a well written manuscript about their hands on learning experience with continuous glucose monitoring. They also have 2 cohorts of data but do not split into two groups to see if learning was similar to combine into one group of data. Missing some key demographic questions for interpretation – number with diabetes, family members with diabetes pre survey (confounders), however that can’t be fixed. They have a large percentage (25%) of students with pre survey good or very good CGM knowledge. It would be interesting to compare learning opinions between the two groups. I would suspect learning would be greater with less pre knowledge. The outcomes are only survey data but they could include test question answers as another good measurement of learning from this hands on experience. A large number (29%) of students dropped from one analysis because CGM fell off before the 14 days, however, they probably had almost similar learning, so you could compare learning with full experience vs less than full experience. Thus the authors could do more analyses to strengthen the outcomes from this learning.
Major
Ln 105 – need more info on the primary survey. Was it for pts, students, providers? Various health care providers? Validated survey? As I can’t tell from the authorship of article was this a company sponsored or independently conducted trial? I need to know how close pharmacy students are to the original intent of the tool. How did you adapt the items? What number of items did you not use? the extracted component? How did you determine what to keep or eliminate? Any reliability or validity testing that you did? This is one of your major endpoints so more details are required.
Ln 117 – sometimes the course is discussed first and then the measurement, however you do the opposite. Might be ok but not the normal presentation mode.
Ln 117 – Nice description of the diabetes component of the course. What was dropped to add 2.5 hrs of CGM material? This is important as one of the number one reasons for not doing this from your intro was lack of time. I think readers will appreciate how you solve this lack of time problem. Why decision to place this training in a therapeutics course vs. your patient care lab?
Ln 150 – as this is a 2.5 hr component of the course, no assessments on the CGM activity were done? Suggest adding the test question data as it would be helpful to understand the learning outcomes of this activity besides just survey data. Understand that the survey was anonymous and thus you can’t pull out the specific student data, but you can note that in the methods that the test outcomes represent the full class. This would be a better outcome of learning than student self-assessment on ability to educate a patient.
Ln 162 – need to describe the process for the qualitative data analysis to develop themes
Ln 162 – will need to add tests for some of the comparisons suggested below.
Ln 188 – An intention to treat analysis would be helpful. 29% is high for number of dropped students because they wore the device for < 14 days. Most of the learning probably comes in the first week with duplication in the second week. So dropping all these students doesn’t seem justified. An intent to treat could solve the problem if learning differed. Or one could check for learning for those with < 14 days to those with 14 days. Add to methods the definition of successful wear used for analysis; as it appears it was for any duration.
Ln 189 – if sensor detached, could they not put on another one? Because of limited supply?
Ln 208 – need to separate CGM understanding from patient counselling abilities – two unique concepts. Not sure how to interpret %s. Is it % of students of % of codes? No methodology on your qualitative analysis so can’t fully interpret. How many students answered each open-ended questions? Insufficient presentation and description in statistical analysis on the qualitative finding process.
Ln 217 – this is not the first article. I did a google search and found other studies. Three other studies have evaluated pharmacy student outcomes, and none of these studies appear in your comparison to current literature in your discussion. Two studies were conducted in a patient care lab, which is part of the core curriculum at least in our college, and one from an elective. One study even used a randomized control study, which is higher power design than a survey. One study used standardized patient assessment, which is also higher power than a survey and even test questions. You will need to add these to your discussions and take off the first study statement.
Curr Pharm Teach Learn 2022 Jan;14(1):62-70. doi: 10.1016/j.cptl.2021.11.021. Epub 2021 Dec 28 Sherril CH et al. , elective
Patient Educ Couns. 2025 Feb:131:108578. doi: 10.1016/j.pec.2024.108578. Epub 2024 Nov 29. (randomized controlled study, with SP assessment; patient care lab) (Folz HN et al)
American Journal of Pharmaceutical Education Volume 87, Issue 8, August 2023, 100312 (patient skills lab) Knezevich E et al
Ln 286 – 303 – biggest limitation is no true assessment of the pharmacy students' ability to educate a person starting or trouble shooting a CGM. Not having a control group is a limitation to actually document true outcomes, however with just opinions your study is partially fine. So it could still be published but the limitation should be included.
Data were not separated by pre knowledge about CGM. 25% of the combined cohort had good to very good knowledge already, which will influence learning and survey responses. It would be interesting for you to compare results between the two groups. With limited demographics, hard to know personal or family experiences with these monitors or the disease state to explain why such high knowledge. Could be a great pharmacy preceptor at their internship also. Did opinions vary by having a pharmacy internship position, and or place of employment vs. those without experience? So you have more you can do to understand the learning experience more.
Minor
Ln 12 – Delete an, doesn’t go with plural surveys
Ln 20 – Add the pe and post % for confident coaching; this is the most important parameter as an outcome measure. Plus without looks like could be small numbers, eg. going from 10% to 20% would be a double
Ln 24 – would add diabetes, pharmacy students, active learning
Ln 33 – this statement needs referencing, which can be done by adding 4 – 7 here
Ln 49 – 55 – usually broad scope of data presented before micro level data – scoping review first and then retrospective cohort study.
Ln 58 and Ln 61 – student pharmacists are doing these education sessions already. They probably should be added to these statements.
Ln 63 – There are about 142 colleges of pharmacy as of 12/23 (AACP), How did you decide to only invite 130 colleges of pharmacy.
Ln 63 - add year of study
Ln 65 – 66 – state as to whether these are elective, patient care labs, or required courses.
Ln 66 – I don’t think you need the quotes
Ln 67 – is this for any education or hands on education?
Ln 70 – you might want to clarify core curriculum to didactic courses as other students have evaluated hands-on CGM learning in elective courses and patient care lab (which is core curriculum)
Ln 80 – Need the outcome of the IRB review, was it reapproved, exempted?
Ln 110 – could say the same Likert scale as above was used.
Ln 129 – add a reference for the video link as a reference like the other 2 sources.
Ln 132 add reference 12 and now the new reference 13 to the activities list – self study table
Ln 146 – spelling change interpenetrations to interpretations
Ln 167 – split out numbers for outpatient and inpatient. Does your outpatient mean community pharmacy or did your outpatient include other types of practices (specialty, clinic, etc)?
Ln 183 – does not applicable also include no answers? Or does the N for each pre-survey item vary? Does nonapplicable mean the student wore a CGM?
Ln 184 – Need to add good between very and knowledge
Ln 165 and 187 – need respond rates by cohort also.
Figure 1 and Figure 2 – why did you have a nonapplicable response? Was it optional or required to apply a CGM? That should also be in the methods. Would be helpful to know what to do if a student does not want to use the monitor.
Ln 228 – 229 – I don’t see the data that supports this statement in your study. As this is not a pt outcome study, this seems to be a stretch and should be removed.
Ln 240 – must be some pharmacy literature on hands on learning outcomes that could be used instead of or in addition to using nursing literature.
Ln 263 – I don’t understand the inevitable statement. Are you saying these devices don’t work well in patients without DM? Could your students have DM, preDM, or hypoglycemic real events for when the monitors go off? What teaching did you provide about alarms for student management? Will they get the full learning experience if the alarms are turned off?
Ln 350 – who is JDRFCGMS? – please spell this out.
Reviewer 6 Report
Comments and Suggestions for Authors
The manuscript aims to present the results of a pre- and post- survey performed among the pharmacy students that participated in diabetes course involving personal experience with continuous glucose monitoring, CGM. The manuscript addresses students’ attitudes towards CGM experience, their understanding of disease as well as their understanding of the educational experience they participated in. The authors provided information regarding feasibility and practicality of introducing the experience into the pharmacy curriculum and assessed its impact on students. The main strength of the manuscript are as follows: clearly presented and well-designed research methodology, clear data presentation, citing new literature. The article is well written, the authors clearly explained the results they obtained, students response rate was high. I would raise some points that could be better addressed by authors:
-
The introduction should be reorganized in order to first define the previous research concerning the education concerning CGM in pharmacy curriculum. Authors did not refer to previous similar studies at all. In fact there have been some previous articles on teaching CGM:
https://accpjournals.onlinelibrary.wiley.com/doi/epdf/10.1002/jac5.2076.
Litten, K., Folz, H., Lobkovich, A., Sherrill, C. H., & Berlie, H. (2025). Integrating continuous glucose monitoring into Doctor of Pharmacy curricula. Journal of the American College of Clinical Pharmacy. J Am Coll Clin Pharm 2025; 8: 129-135
https://www.ajpe.org/article/S0002-9459(24)10781-4/abstract.
Blake, E. W., Dunn, B., Smith, G., Clements, J., Sides, A., & Keisler, B. D. (2024). SWEET-DM-Students Wearing External Glucose Monitors to Educate and Treat Patients with Diabetes Mellitus. American Journal of Pharmaceutical Education, 88(9).
Moreover some authors also let students use CGM on them in order to better understand hurdles of disease management an monitoring (https://doi.org/10.1016/j.pec.2024.108578 and https://doi.org/10.1016/j.cptl.2021.11.021) . Introduction should be based on those and similar articles, present the research area and define a research gap. At present the manuscript is formulated in a way as if this was the very first approach to the use of CGM by students during the course.
- Accordingly, the discussion should be rewritten in order to include comparing the results of authors’ results with previous, abovementioned studies. The reader should know how this new research fits into the whole development of the use of CGM experience as teaching and learning methodology.
- The authors should discuss the acceptance of CGM among students and relatively high detachment rate (as compared to general patient population) addressing the previous studies on youth acceptance of CGM, e.g. https://doi.org/10.32873/unmc.dc.ihsej.0041,. This would shed more light onto perspective of students that did not complete the whole two week wear.
- It would also be interesting to compare the data of students (attitude towards the novel course design and towards their own health status) in both groups i.e. students who completed the entire CGM experience vs those who did not.
I would also highlight some minor points:
- In line 125 ‘CDES’ abbreviation should be explained
Round 2
Reviewer 4 Report
Comments and Suggestions for Authors
I appreciate the authors’ efforts to revise the manuscript and address prior feedback. However, after reviewing the revised submission, I find that my initial recommendation for rejection remains. While the topic of CGM training is timely and potentially valuable for pharmacy education, there are still significant concerns related to study design, interpretation of results, and alignment between the stated objectives and the data collected. The revisions made do not sufficiently resolve earlier issues, and several key concerns persist. Below, I outline specific points to support further refinement of the work, should the authors choose to further revise and/or resubmit:
-The abstract states: “The number of students who acknowledged the importance of CGM training increased from 56% to 94% after receiving the CGM training.” However, the pre- and post-survey items differ in wording and emphasis. The pre-survey asks whether CGM training should be mandatory in PharmD education, while the post-survey addresses agreement with not recommending CGM sensor wear in the curriculum. These items are not directly comparable, and framing the change as an increase in perceived importance is potentially misleading. Furthermore, the conclusion that CGM training should be mandated in the curriculum is not clearly supported by the data.
-The abstract also states: “The number of students who felt confident coaching patients on CGM use more than doubled from 30% to 73%.” It is unclear which survey items this is based on. Confidence in coaching patients does not appear to be assessed directly in the pre-survey, and there is inconsistency in reported figures between the abstract (73%) and the discussion (85%). These discrepancies make interpretation difficult and weaken the credibility of the reported outcomes. The 85% number also leads me to believe that this refers to the statement about students feeling confident talking to their patients and peers about CGM. If this is correct, it is critical to understand that talking to peer and coaching patients are different tasks.
-The authors state that “the purpose of this study was to evaluate the sentiments of how the students felt about having hands-on CGM education as part of their core curriculum,” however the pre/post survey items do not assess this study purpose/objective. This study objective is not indicated in the main text of the manuscript, only in the abstract, which contributes to lack of clarity about the aims of the study. This was noted in the initial peer review. The authors responded that “The intention was to incorporate the hands-on CGM training into the core curriculum to benefit all student pharmacists and help them become practice-ready.” The stated study purpose above does not focus on assessing practice-readiness. Additionally, the authors indicated that the students’ user experiences were solicited in alignment with the study objective of examining students’ perceptions of receiving hands-on CGM training as part of their core therapeutics. Perceptions of having a training experience within core pharmacy curricula and perception of comfort of the CGM device are unrelated objectives. It could be possible that understanding the students’ user experience and benefits to their own health are secondary objectives; however, this is not stated in the manuscript. I recommend the authors create clear study objectives, include these in the manuscript text, and align presented results and conclusions with these objectives.
-Some helpful clarifications included in the author response (e.g., Response 6) would benefit the manuscript if incorporated directly. Additionally, Response 8 notes an effort to avoid duplicating results between figures and text; however, this change is not readily apparent in the revised manuscript. Continued revisions in this area would improve readability.
-The assertion that CGM training improved students’ overall understanding of diabetes is based solely on perception-based survey responses. While it is valid to report students’ self-reported confidence or comfort, drawing conclusions about broader understanding may be premature without additional assessment tools.
Reviewer 5 Report
Comments and Suggestions for Authors
The authors have responded to all the queries. Most have resulted in some changes, or explanations to the reviewer that do not appear in the text. I suspect some of the comments to the reviewer would still be of interest to the readers.
Ln 168 – 170 the details on the qualitative data analysis provide who did the analysis and that themes were sought but the process of determining the themes is not described. As the major component of this paper is qualitative analysis, this should be described in greater detail.
Figure 1 and 2 The not applicable responses for examples “I feel nervous about placing a sensor on my arm.” and “Wearing glucose sensor made it harder for me to sleep.” I now understand was a Likert scale response, but they still don’t make sense on why that was even an option. But because it was a response it does need to be included.
I understand the authors do not want to use the test question answers as part of the learning from this hands-on experience, but they could have downloaded the answers to these questions as a deidentified database and used without student assess/consent. Using this deidentified database would still qualify as exempted SOTL.
Limitations – I still think a limitation is that previous knowledge and personal use were not controlled for/used as a confounder is important to include. Yes you do have positive results, but different subgroups could have greater or less learning from this hands on experience.
For the future studies, you should include to determine how well this hands-on experience relates to improved patient counseling of these devices; one of the major downstream reasons for learning this material.
Author Response
Thank you for the feedback. Please see attached.

Reviewer 6 Report
Comments and Suggestions for Authors
As authors referred to my previous comments, I believe the manuscript is now ready for publication.
Author Response
Thank you for your acknowledgement and support.